# Spatiotemporal Variation and Factors Influencing Water Yield Services in the Hengduan Mountains, China

Qiufang Shao [1], Longbin Han [2], Lingfeng Lv [2], Huaiyong Shao [2,3,*] and Jiaguo Qi [4]

[1] Teaching Steering Committee, Sichuan Tourism University, Chengdu 610100, China; 0001318@sctu.edu.cn
[2] College of Earth Sciences, Chengdu University of Technology, Chengdu 610059, China; hlb@stu.cdut.edu.cn (L.H.); lvlingfeng@stu.cdut.edu.cn (L.L.)
[3] Key Laboratory of Earth Exploration and Information Technology, Ministry of Education, Chengdu 610059, China
[4] Center for Global Change and Earth Observations, Michigan State University, East Lansing, MI 48824, USA; qi@msu.edu
* Correspondence: shaohuaiyong@cdut.edu.cn

**Abstract:** Conducting a quantitative assessment of water yield in mountainous areas is crucial for the management, development, and sustainable utilization of water resources. The Hengduan Mountains Region (HDMR) is a significant water-supporting area characterized by complex topography and climate changes. To analyze the spatial and temporal variations of water yield in the HDMR from 2001 to 2020, we employed the InVEST model and examined the influencing factors in conjunction with the elevation gradient. Our results indicate that: (1) The water yield in the Hengduan Mountains decreases from southeast to northwest, with the southwestern and eastern regions having high water yield values, and the high-altitude areas in the northwestern part having low water yield values. (2) The water yield in the Hengduan Mountains exhibits a decreasing trend followed by an increasing trend from 2001 to 2020, with the lowest level in 2011 and higher levels in 2004, 2018, and 2020. (3) Pixel-based trend analysis demonstrates a decreasing trend in water yield in the central and western parts of the study area, while the eastern part shows an increasing trend. (4) The climatic components, particularly precipitation, predominantly influence the spatial and temporal variations of water yield in the Transverse Mountain region. In most areas, evapotranspiration and land surface temperature have a negative impact on water yield. (5) Water yield tends to decrease and then increase on the altitudinal gradient, with precipitation and actual evapotranspiration being the factors directly affecting water yield, and land surface temperature and the proportion of forested areas having a significant indirect effect on water yield. Our study provides a scientific basis for water resources management and sustainable development in the Hengduan Mountains.

**Keywords:** ecosystem services; water yield; InVEST model; Hengduan Mountains; gradients





## 1. Introduction

Ecosystem services are the benefits that humans derive from ecosystems and can be categorized into four types: provisioning (e.g., seafood and timber), regulation (e.g., climate and flooding), support (e.g., food production and pest control pollination), and culture (e.g., tranquility and inspiration) [1,2]. Water yield services, which are part of the hydrological cycle, play a crucial role in the exchange of the Earth's energy and the transfer of chemicals. They also play a significant role in water purification, flood control, and runoff regulation [3,4]. As an important ecosystem service [5], water yield services are vital for ecosystem balance and regional sustainable development.

In the realm of assessing ecosystem water services, several methods have been employed, each with its own unique characteristics and complexities. These methods include water balance, precipitation storage [6], integrated water storage, soil storage, canopy retention residual, and multi-factor regression, which collectively contribute to a better

understanding of ecosystem water services [6]. Among these methods, the Integrated Valuation of Ecosystem Services and Tradeoffs (InVEST) annual water yield module is based on the water balance approach [7]. This approach defines water yield as the amount of precipitation that remains after evapotranspiration and groundwater recharge have occurred. The InVEST model has been extensively employed for watershed water yield simulation [4,8–11], and its reliability in data-scarce regions has been validated [12]. These studies have underscored the robustness and versatility of the InVEST water yield model, which has been effectively applied in regional water resource assessment.

Mountain ecosystems, which are dominated by grasslands and forests, cover almost a quarter of the Earth's land area. About half of the world's mountains provide downstream water resources, making them critical water-bearing regions [13,14]. However, the threats to mountain ecosystem services are increasing due to climate change and human activities [15]. Studies have demonstrated that rapid population growth has resulted in water shortages in most mountain ranges worldwide [16]. Therefore, managing water resources in mountains poses a significant challenge.

The current research regarding factors that affect water yield services in mountainous regions has primarily focused on climate change and human activities. Changes in precipitation and evapotranspiration due to climate change can considerably influence the regional water cycle, thus impacting water production. Furthermore, it is essential to acknowledge that human-induced alterations in vegetation and land use can also significantly impact water yield services. Vegetation is vital in the watershed water cycle process by regulating hydrological processes such as evapotranspiration, surface run-off, and soil infiltration [17], thereby affecting energy and water balance [18,19]. The ecohydrological processes related to climate and vegetation at different altitudes behave differently [20–24], resulting in changes in water yield along the altitudinal gradient. Currently, many scholars have introduced the effect of terrain differences to carry out related research, such as when Zhang introduced the Terrain Niche Index to analyze the impact of the terrain gradient on ecosystem services in the Heihe River Basin, and pointed out that there is a positive correlation between water yield and the Terrain Niche Index [25], but Ma et al. pointed out that water yield was relatively low in the high topographic gradient zone, and there was a decreasing trend of water yield with an increasing elevation [26,27]. Apparently, there are regional variations in water yield with the altitude gradient, which are related to the climate, vegetation and land use types at different altitude gradients. Fewer studies have been conducted to quantify the direct and indirect effects of factors on water yield under the altitudinal gradient, and further clarification is needed.

The Hengduan Mountains are located in southwest China, with complex mountainous topography and significant vertical climate changes [28]. They possess the most characteristic complex elevation zones in Asia and Europe [29] and are also the upstream areas of several rivers, such as the Jinsha, Yalong, and Lancang, which are vital for water conservation [30,31]. Unfortunately, human activities, including vegetation destruction, overgrazing, and wasteland reclamation, have seriously damaged the local ecosystem, making it one of the most eroded areas in China [32–34]. Although previous studies of water yield in the Hengduan Mountains have focused on climatic factors and land use and land cover (LULC) changes [30,35,36], the complex topography of the Hengduan Mountains also requires consideration of the impact of topographic gradients on the spatial and temporal changes in water yield services. Therefore, this study employs the InVEST model to calculate water yield in the Transverse Mountain region over the past 20 years and explores its spatial and temporal variation characteristics. The study also investigates the effects of climate, vegetation, and LULC changes on water yield services under different topographic gradients, providing valuable insights into water security management measures in mountainous areas.

## 2. Study Area and Data Sources

### 2.1. Study Area

The Hengduan Mountains Region (HDMR) covers a total area of about 49,500 km$^2$ and is located in southwestern China, including western Sichuan Province, northern Yunnan Province, and eastern Tibet Autonomous Region (Figure 1). The area exhibits a complex topography, with an average elevation exceeding 3000 m and decreasing from northwest to southeast. Precipitation also varies significantly with altitude, as the region is located in the transition zone of the first and second terraces in China. The region boasts diverse vegetation types, including coniferous forests, shrubs, grasslands, meadows, mixed coniferous forests, broadleaf forests, bogs, alpine vegetation, and artificial vegetation. Moreover, the region is intersected by several rivers such as the Nujiang River, Jinsha River, Yalong River, and Lancang River, thereby making it a significant water-conserving area in China.

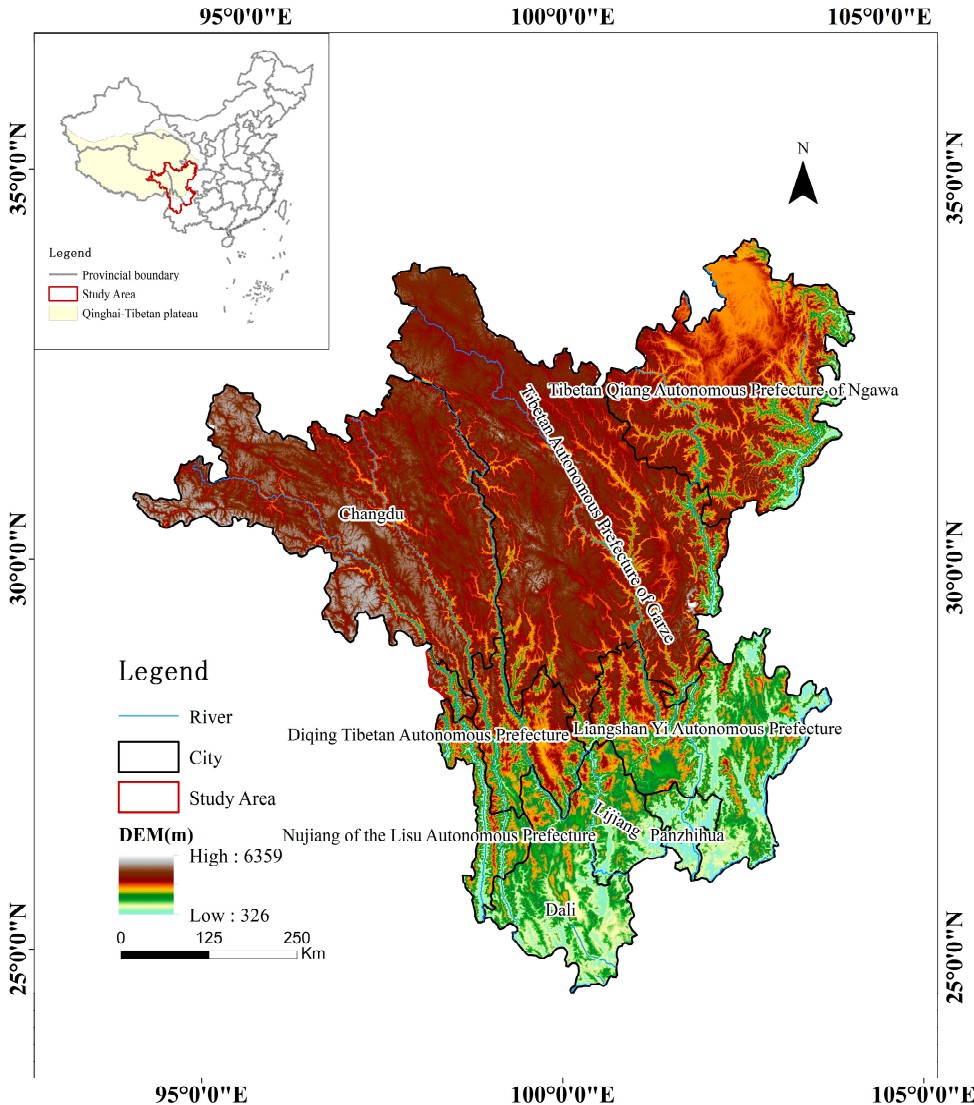

**Figure 1.** Geographical location and elevation of the HDMR.

### 2.2. Data Source and Processing

The primary data sources utilized in this paper encompass multiple facets of the natural environment. The required data sources and presentations are shown in Table 1 and the LULC coefficients are tabulated in Table 2. All the data is resampled to a 500 m resolution. Table 2 shows the LULC coefficients, reclassifying the LULC into six categories. lulc_veg

is assigned a value of 1, according to the InVEST handbook for vegetation cover land use types other than wetlands, and a value of 0 for all other land use types. The root_depth represents the maximum rooting depth, and Kc represents the evapotranspiration coefficient for each LULC type, with values referencing the sources of reference to previous studies as well as to the InVEST model's recommended parameters [37–40]. Multiple simulations were performed and a Z value of 15 was determined when the watershed yield and the total water yield from the water resources bulletin were closest.

**Table 1.** Data Description.

| Data Types | Resolution | Period | Data Sources | Brief Introduction |
|---|---|---|---|---|
| Land use/land cover (LULC) | 500 m | 2001–2020 | MODIS MCD12Q1, Terrestrial Process Distributed Activity Archiving Center (LP DAAC) (https://lpdaac.usgs.gov/, accessed on 9 February 2023) | The data was reclassified into six categories based on Table 2. |
| Digital elevation model (DEM) | 500 m | - | Resources and Environmental Sciences and Data Center, Chinese Academy of Sciences (https://www.resdc.cn/, accessed on 9 February 2023) | Radar topographic mapping SRTM derived from the U.S. Space Shuttle Endeavour. |
| Precipitation (PR) | 1/24° | 2001–2020 | Monthly climate and climate–water balance datasets on the global land surface, TerraClimate (https://www.nature.com/, accessed on 9 February 2023) | Monthly values were synthesized into annual precipitation data for further analysis |
| Reference evapotranspiration (ET0) | 1/24° | 2001–2020 | Monthly climate and climate–water balance datasets on the global land surface, TerraClimate (https://www.nature.com/, accessed on 9 February 2023) | Monthly values were synthesized into annual reference evapotranspiration data for further analysis |
| Land surface temperature (LST) | 1000 m | 2001–2020 | MODIS MOD11A2, Terrestrial Process Distributed Activity Archiving Center (LP DAAC) (https://lpdaac.usgs.gov/, accessed on 9 February 2023) | |
| Soil | 1000 m | - | Soil dataset of China at the Harmonized World Soil Database (HWSD) (v1.1) (2009) (http://poles.tpdc.ac.cn/, accessed on 9 February 2023) | Involves maximum soil root depth (mm), clay content (%), powder content (%), sand content (%), organic matter content (%), etc. |
| NDVI | 500 m | 2001–2020 | MODIS MOD13A1, Terrestrial Process Distributed Activity Archiving Center (LP DAAC) (https://lpdaac.usgs.gov/, accessed on 9 February 2023) | The maximum value of annual. NDVI was obtained using the maximum value synthesis method (MVC) after removing outliers. |
| Watershed | (Vector data) | - | Sciences and Data Center, Chinese Academy of Sciences (https://www.resdc.cn/, accessed on 9 February 2023) | Includes all river networks in the country and all sub-basins with an area greater than 100 km$^2$ |
| Actual total water yield | - | - | Water Resources Bulletin | For verification of water yield |

**Table 2.** Table of LULC coefficients.

| Primary Classification | Secondary Classification | Lulc_veg | Root_depth | Kc |
|---|---|---|---|---|
| forest | Evergreen Needleleaf Forests<br>Evergreen Broadleaf Forests:<br>Closed Shrublands<br>Open Shrublands<br>Mixed Forests<br>Deciduous Needleleaf Forests<br>Deciduous Broadleaf Forests | 1 | 5000 | 0.9 |
| grass land | Woody Savannas<br>Savannas<br>Grasslands | 1 | 600 | 0.65 |
| farm land | Croplands<br>Cropland/Natural Vegetation Mosaics | 1 | 500 | 0.65 |
| waterbody | Permanent Wetlands<br>Permanent Snow and Ice<br>Water Bodies | 0 | 1 | 1 |
| construction land | Urban and Built-up Lands | 0 | 1 | 0.3 |
| unused land | Barren | 0 | 1 | 0.25 |

## 3. Research Methods

### 3.1. Water Yield Model

The InVEST water yield module calculates the water yield of an image element based on the water balance method and the Budyko water–heat coupling equilibrium assumption, as shown below:

$$Y(x) = \left(1 - \frac{AET(x)}{P(x)}\right) \cdot P(x) \tag{1}$$

In Equation (1), *AET* denotes the annual actual evapotranspiration (mm), and *P* denotes the annual precipitation (mm). $\frac{AET(x)}{P(x)}$ is calculated using the Budyko water–heat coupled equilibrium assumption equation proposed by Fu [41] and Zhang [42], as shown below:

$$\frac{AET(x)}{P(x)} = 1 + \frac{PET(x)}{P(x)} - \left[1 + \left(\frac{PET(x)}{P(x)}\right)^{\omega}\right]^{1/\omega} \tag{2}$$

In Equation (2), *PET* denotes potential evapotranspiration (mm), and ω denotes the non-physical parameter of natural climate–soil properties, which is calculated as follows:

$$PET(x) = K_c(l_x) \cdot ET_0(x), \omega(x) = Z\frac{AWC(x)}{P(x)} + 1.25 \tag{3}$$

In this equation, $ET_0$ represents the reference evapotranspiration, and $K_c$ is the plant or vegetation evapotranspiration coefficient for a specific land use/land cover (LULC) type. The coefficient *Z* is the Zhang coefficient, and *AWC* (*x*) is the available water content of plants, which is calculated as follows:

$$AWC(x) = Min(\text{Rest. layer. depth, root. depth}) \cdot PAWC \tag{4}$$

*PAWC* indicates the plant available water content and can be derived by calculating the difference between the field water holding capacity and the permanent wilting coefficient [43], which is calculated as follows:

$$FMC = 0.003075 \times \text{Sand} + 0.005886 \times \text{Slit} + 0.008039 \times \text{Clay} + 0.002208 \times 0M - 0.14340 \times \text{BD} \tag{5}$$

$$WC = -0.000059 \times \text{Sand} + 0.001142 \times \text{Silt} + 0.005766 \times \text{Clay} + 0.002228 \times 0M + 0.02671 \times BD \tag{6}$$

In Equations (5) and (6): *FMC* is the field water holding capacity; *WC* is the permanent wilting coefficient; Clay, Silt, Sand, and *OM* are the constituents of soil, and *BD* refers to the bulk density of the soil.

### 3.2. Trend Analysis and Testing

The Theil–Sen Median (Sens) slope estimation method for calculating the slope of change of a long time series is a nonparametric method proposed by Sen in 1968 for evaluating the trend of sample data points [44]. Its calculation formula is as follows:

$$TS_{slope} = \text{median}\left(\frac{x_j - x_i}{t_j - t_i}\right), \forall t_j > t_i \tag{7}$$

In Equation (7), $TS_{slope}$ denotes the median function, and $x_j$ and $x_i$ denotes the value of the jth and ith year. If $TS_{slope}$ is greater than 0, it means that the data has an increasing trend; if $TS_{slope}$ is less than 0, it means that the data has a decreasing trend; and its absolute value indicates the size of the trend change. The Mann–Kendall (MK) test is a nonparametric test, the advantage of which is that it does not require the sample to follow a certain distribution law and can effectively exclude the interference of outliers. For the time series X, the MK trend test statistic is:

$$Z_c \begin{cases} \frac{S-1}{\sqrt{Var(S)}}, S > 0 \\ 0, S = 0 \\ \frac{S+1}{\sqrt{Var(S)}}, S < 0 \end{cases}, \quad S = \sum_{i=1}^{n-1}\sum_{j=i+1}^{n} sgn(x_j - x_i) \tag{8}$$

$$sgn(\theta)\begin{cases} 1, \theta > 0 \\ 0, \theta = 0 \\ -1, \theta < 0 \end{cases}, Var(S) = \frac{n(n-1)(2n+5) - \sum_{i=1}^{n} t_i(i-1)(2i+5)}{18} \tag{9}$$

## 4. Results

### 4.1. Spatial Pattern of HDMR Water Yield

The spatial distribution of water yield in the HDMR from 2001 to 2020 exhibited obvious spatial heterogeneity, with a general decrease from southeast to northwest (Figure 2). The average annual water yield (AWY) in most areas over the 20-year period was concentrated between 300 mm and 700 mm, accounting for about 95% of the area. Notably, the high-value region was mainly clustered in the southwest and eastern sectors of the HDMR, including Lushui, Fugong, and Yunlong counties in western Yunnan Province, as well as Songpan, Heshui, and Lixian counties, among others, in central and western Sichuan Province, with some areas showing an average AWY greater than 600 mm and even greater than 1000 mm. Conversely, the average water yield in eastern Tibet was lower, particularly in Changdu, Chaya, Mankang, Yanjing, and Batang counties, with the AWY in some areas less than 400 mm.

### 4.2. Trend Analysis of HDMR Water Yield

Between 2001 and 2020, water yield showed a decreasing and then increasing trend, reaching its lowest value in 2011, when the average water yield dropped to 406 mm (Figure 3).

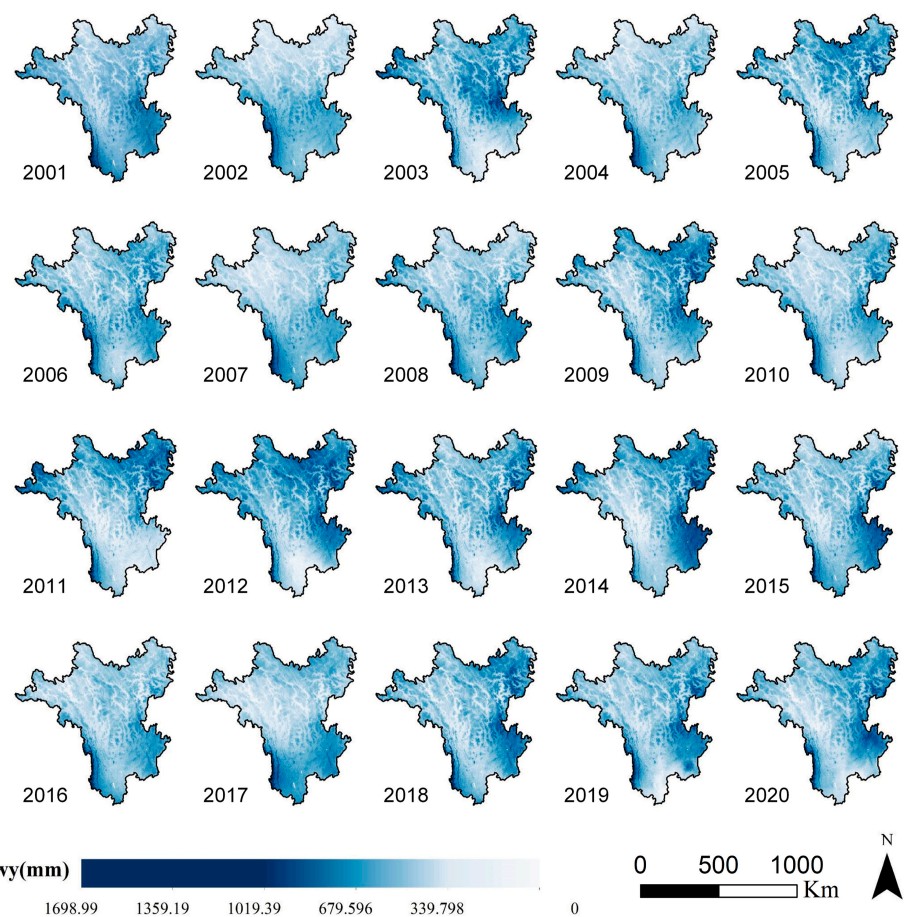

**Figure 2.** The HDMR water yield from 2001 to 2020.

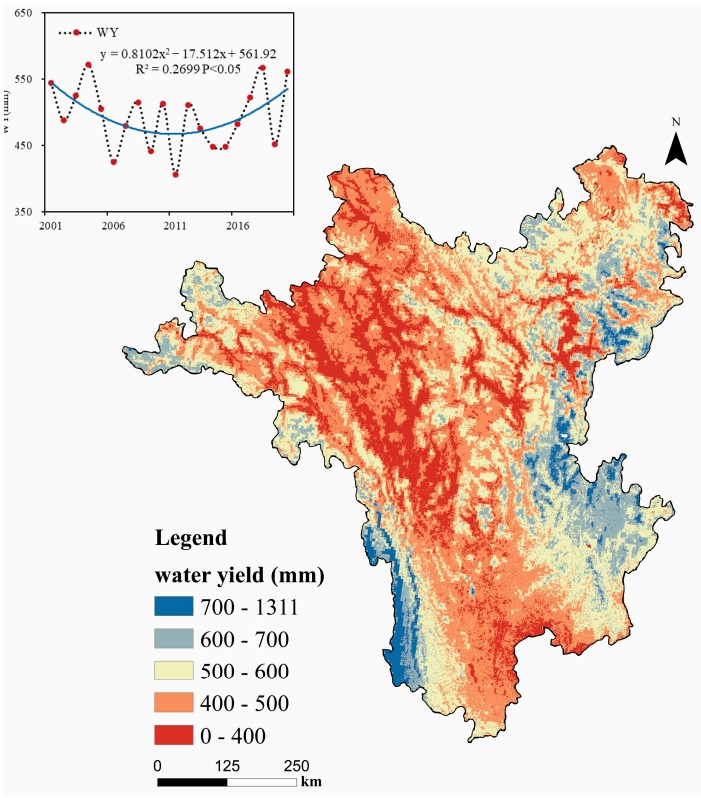

**Figure 3.** Temporal variation in the annual average water yield of the HDMR.

Figure 4 demonstrates that it is feasible to gauge the trend of WY from 2001 to 2020 by employing the Sens trend analysis and MK test. The spatial distribution of the water yield variation trend evidently differs, with the water yield variation region mainly concentrating in the central and western regions of the HDMR and the eastern Sichuan Basin region. Within these areas, the water yield trend in the west and central regions is on a downward trajectory. Some regions had a slope exceeding −40 mm/a, with the percentage of the area of slightly significant decrease, significant decrease, and highly significant decrease amounting to 5.52%, 6.83%, and 0.37% (Table 3), respectively, and was mainly concentrated in Zogong County, Yanjing County, Xiangcheng County, and Litang County. The percentage of the area of slightly significant increase, significant increase, and highly significant increase amounted to 5.16%, 2.32%, and 0.15%, respectively, and was mainly concentrated in Mao County, Wenchuan County, Meigu County, and Zhaoge County. The highest slope noted was 45.15 mm/a, and 79.61% of the area exhibited no significant change in water yield.

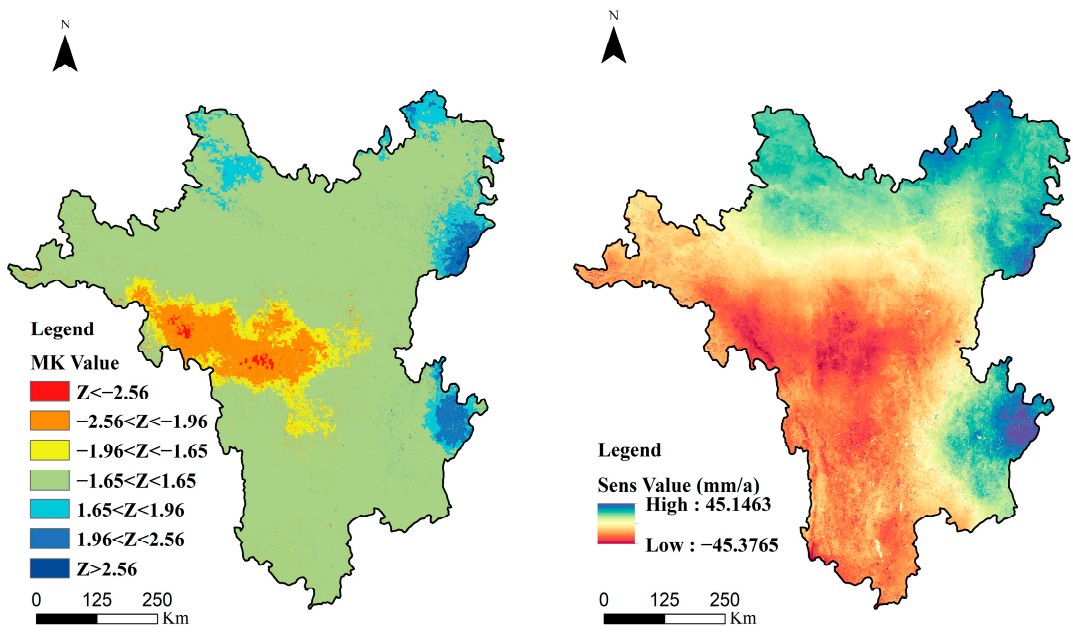

**Figure 4.** Results of the MK test and SENS trend analysis.

**Table 3.** Area and percentage of MK value.

| MK Value | Area (km$^2$) | Percentage (%) |
|---|---|---|
| Z < −2.56 | 1851.972131 | 0.37 |
| −2.56 < Z < −1.96 | 33,856.09 | 6.83 |
| −1.96 < Z < −1.65 | 27,356.32 | 5.52 |
| −1.65 < Z < 1.65 | 394,143.04 | 79.61 |
| 1.65 < Z < 1.96 | 25,559.08 | 5.16 |
| 1.96 < Z < 2.56 | 11,525.34 | 2.32 |
| Z > 2.56 | 783.72 | 0.15 |

### 4.3. Factors Influencing Water Yield

4.3.1. The Influence of Climatic Factors on Water Yield

The spatial distribution patterns of precipitation PR and water yield WY for the past two decades in the HDMR show that the high PR area is located in the western part of Yunnan Province (Figure 5), mainly influenced by southwest water vapor [45], while the southern part of Sichuan Province is also characterized by higher PR, mainly controlled by the southeast monsoon. However, the northwest of HDMR has relatively low annual average PR due to the obstruction of the north–south mountain ranges and the increase in altitude. The northern valley area of HDMR experiences less precipitation compared to the

surrounding areas due to the rain shadow or the Föhn effect on the leeward slope of tall mountains [46]. Combined with a higher potential evapotranspiration (PET), it results in a lower regional WY. PET, actual evapotranspiration (AET), and land surface temperature (LST) show a similar spatial distribution pattern, with higher AET and surface temperature in low-altitude areas in the south that decreases from south to north.

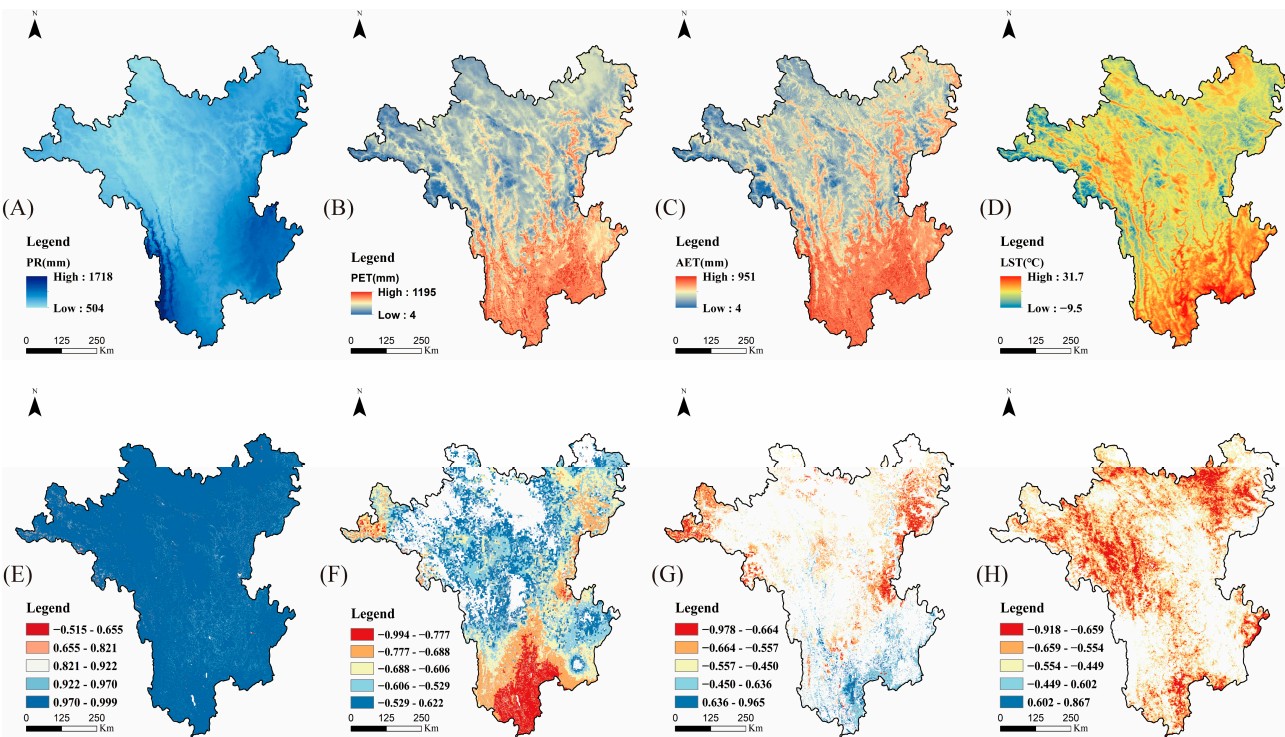

**Figure 5.** (**A**–**D**) Represents the 20-year average PR, PET, AET, and LST; (**E**–**H**) shows their Poisson correlation coefficients with WY ($p < 0.05$).

The temporal evolution of PR in the HDMR is similar to that of water yield, with a decreasing and then increasing trend from 2001 to 2020, reaching a minimum in 2011 which coincided with other findings [47]. The year 2011 was characterized by exceptional high temperatures and scarce precipitation, which triggered an acute drought event [48], leading to the lowest WY in the past two decades. Similarly, 2006 had low PR averages and relatively high PET levels, resulting in a massive drought episode in the eastern and Sichuan regions of HDMR, which are also reflected in Figure 2. Although PET, AET, and LST showed a weak upward trend from 2001 to 2020, the trend was not significant (Figure 6).

The correlation analysis between WY and PR, PET, AET, and LST showed that al-most all regions in the HDMR had a correlation coefficient of over 90% between PR and WY (Figure 5). In 69.1% of the regions, PET had a negative correlation with WY with an average correlation coefficient (R) of −0.61, which was mainly concentrated in the low-altitude areas in northern Yunnan Province. In 19.5% of the regions, AET had a negative correlation with WY, with an average R of −0.58. However, in some low-altitude areas in southern HDMR, WY showed a positive correlation with AET, which may be related to higher precipitation. In 59.8% of the regions, LST had a negative correlation with WY, with an average R of −0.57. These findings are consistent with previous research results [49,50], suggesting that precipitation, as the source of water yield, is the main variable affecting the spatiotemporal pattern of water yield.

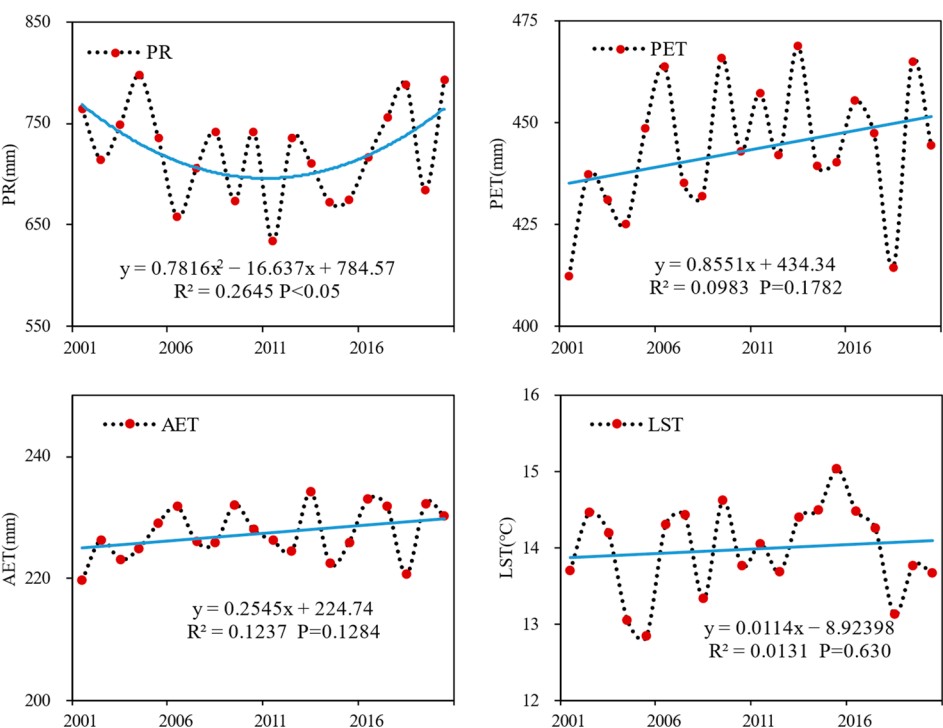

**Figure 6.** Annual average PR, PET, AET, and LST.

#### 4.3.2. The Influence of LULC on Water Yield

Figure 7 shows the average WY of different LULC types in the HDMR from 2001 to 2020. The results indicate that WY of different LULC types changed similarly over the study period. The highest average WY was on unused land, reaching a maximum of 715 mm in 2004. The average WY of construction land and cultivated land reached their peaks in 2017, at 658 mm and 669 mm, respectively. Grassland and forest had relatively high average WY in 2004, 2018, and 2020. From 2002 to 2004, the average WY of water bodies, grasslands, and unused land continued to increase, while the average WY of forest land, construction land, and cultivated land first decreased and then increased, reaching a low point in 2003.

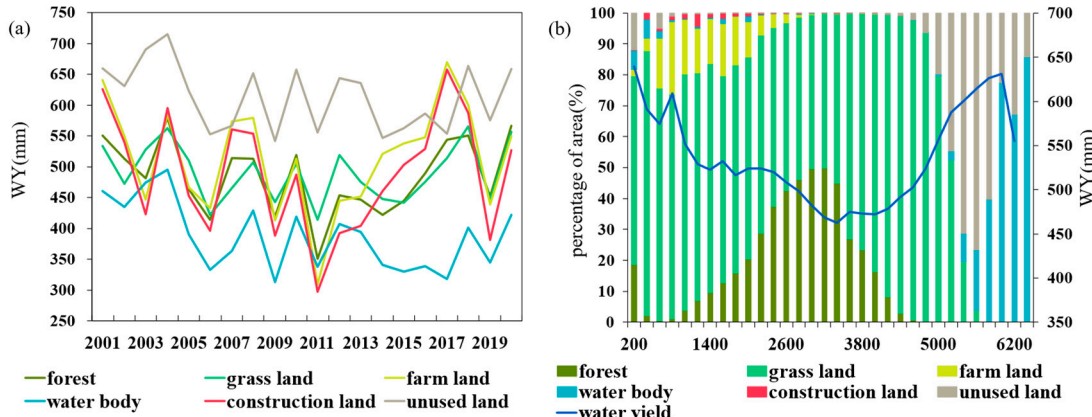

**Figure 7.** (**a**) Shows the average WY change of each LULC from 2001 to 2020; and (**b**) shows the percentage of LULC at different elevation gradients and WY.

The LULC in the HDMR are distributed at varying altitudes, with cropland and built-up areas concentrated at low altitudes, and water bodies (mainly wetlands) and unused land at high altitudes (>5000 m) (Figure 8). The proportion of LULC at different altitudes was calculated in 200-meter intervals (Figure 7). The proportion of forest cover increases

with an increasing altitude and then gradually decreases, reaching its peak (49.8%) at around 3200 m. Before 5000 m, the proportion of grassland decreases, in contrast to forests, then increases and reaches its lowest value (49.7%) at around 3200 m, and its highest value (96.9%) at around 4600 m. After 5000 m, the proportion of grassland decreases rapidly. Forests in the mid to high-altitude areas are mainly distributed around river valleys. The proportion of different LULC types changed very little from 2001 to 2020, with forest and farmland being the main types (Table 4). The proportion of forest increased from 16.32% in 2001 to 17.9% in 2020, while the proportion of farmland decreased from 2.2% in 2001 to 1.6% in 2020, which may be related to the government's policy of returning farmland to forests.

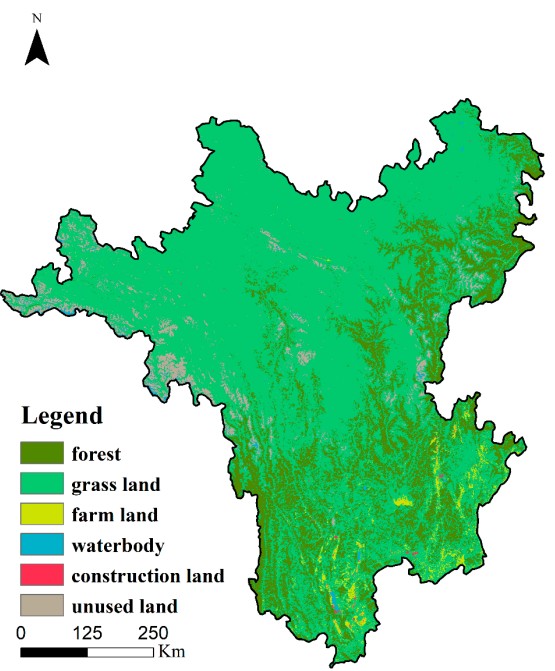

**Figure 8.** The distribution of LULC in the HDMR in 2020.

**Table 4.** Percentage of LULC by year 2001–2020 (Unit: %).

|  | Forest | Grass Land | Farm Land | Water | Construction Land | Unused Land |
|---|---|---|---|---|---|---|
| 2001 | 16.33 | 76.46 | 2.22 | 0.52 | 0.13 | 4.35 |
| 2002 | 16.29 | 76.60 | 2.26 | 0.45 | 0.13 | 4.27 |
| 2003 | 16.26 | 76.65 | 2.29 | 0.41 | 0.13 | 4.26 |
| 2004 | 16.36 | 76.61 | 2.27 | 0.41 | 0.13 | 4.23 |
| 2005 | 16.47 | 76.63 | 2.23 | 0.42 | 0.13 | 4.12 |
| 2006 | 16.55 | 76.68 | 2.20 | 0.39 | 0.13 | 4.05 |
| 2007 | 16.69 | 76.62 | 2.18 | 0.38 | 0.13 | 4.01 |
| 2008 | 16.69 | 76.63 | 2.19 | 0.38 | 0.13 | 3.99 |
| 2009 | 16.65 | 76.75 | 2.18 | 0.36 | 0.13 | 3.94 |
| 2010 | 16.53 | 76.88 | 2.16 | 0.37 | 0.13 | 3.92 |
| 2011 | 16.57 | 76.95 | 2.10 | 0.39 | 0.13 | 3.86 |
| 2012 | 16.50 | 77.09 | 2.07 | 0.39 | 0.13 | 3.83 |
| 2013 | 16.67 | 77.01 | 2.01 | 0.40 | 0.13 | 3.79 |
| 2014 | 16.74 | 76.95 | 1.96 | 0.40 | 0.13 | 3.82 |
| 2015 | 16.92 | 76.85 | 1.91 | 0.39 | 0.13 | 3.81 |
| 2016 | 17.12 | 76.67 | 1.89 | 0.40 | 0.13 | 3.80 |
| 2017 | 17.45 | 76.30 | 1.83 | 0.40 | 0.13 | 3.89 |
| 2018 | 17.42 | 76.04 | 1.77 | 0.46 | 0.13 | 4.18 |
| 2019 | 17.51 | 76.15 | 1.69 | 0.50 | 0.13 | 4.02 |
| 2020 | 17.92 | 76.00 | 1.62 | 0.49 | 0.14 | 3.83 |

The WY curve shows a trend of first decreasing and then increasing, with the lowest point (468.8 mm) around 3000 m. This trend may be related to the percentages of grassland, forest, and unused land. Unused land has low vegetation coverage, weak evaporation capacity, and surface permeability [51,52], resulting in a higher WY. Forest, on the other hand, has deeper roots and stronger permeability, but requires more water and has a stronger evaporation capacity [53–55]. The PET and actual evapotranspiration AET in the forest region are both high, which leads to a lower WY in the forest. Considering that the HDMR area is the origin of many rivers, and its WY service affects the water security and sustainable development of downstream areas, afforestation projects should also consider their impact on the watershed water resources.

## 5. Discussion

### 5.1. Verification of InVEST

The study compared the model predictions with the measured data and continuously adjusted the model parameters to obtain the best fit values for the study area. Utilizing the water resources bulletin of Sichuan and Yunnan Provinces in 2019, Figure 2 shows the simulated water yield and actual total water yield of eight cities: Diqing Tibetan Autonomous Prefecture, Nujiang of the Lisu Autonomous Prefecture, Dali, Lijiang, Panzhihua, Liangshan Yi Autonomous Prefecture, Tibetan Autonomous Prefecture of Garze, and Tibetan Qiang Autonomous Prefecture of Nga-wa. The results show that there is a strong linear relationship between the two (Figure 9), indicating that the InVEST model is effective in simulating water yield.

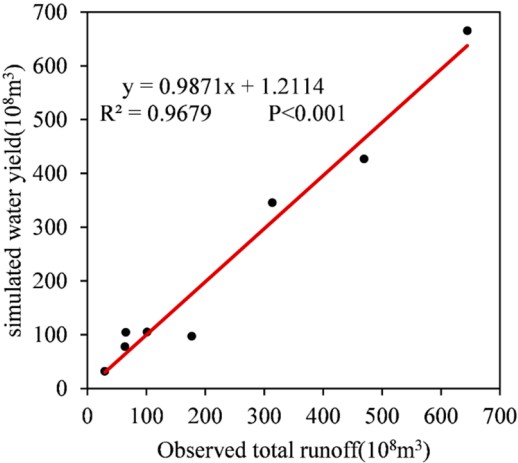

**Figure 9.** Water yield verification.

### 5.2. Effect of Vegetation Cover on WY

WY is influenced by multiple factors, including vegetation coverage, soil type, and vegetation type. Figure 10 shows the distribution of NDVI in the HDMR area. To investigate the relationship between vegetation coverage and water yield at different elevations, the HDMR was divided into four altitudinal zones based on the classification by Long et al. [56]: low altitude (below 1000 m), middle altitude (between 1000 m and 3500 m), high altitude (between 3500 m and 5000 m), and very high altitude (above 5000 m). The WY values at different elevations in different years were calculated by excluding areas classified as water bodies and construction land (Figure 11). The WY was strongly positively correlated with the NDVI in the low-altitude interval ($R^2 = 0.9511$, $p < 0.001$). For every 0.1 increase in the NDVI, WY increased by 59.84 mm. In the middle-altitude interval where the NDVI was less than 0.65, water yield had a weak positive correlation with the NDVI ($R^2 = 0.721$, $p < 0.001$) and did not vary significantly with the NDVI, but when the NDVI was greater than 0.65, water yield increased with the NDVI. At high and very high altitudes, WY decreased with the increasing NDVI. These results indicate that the relationship between the NDVI and

WY is not simply linear and varies significantly with elevation, with a positive correlation at low and medium elevations and a negative correlation at high elevations.

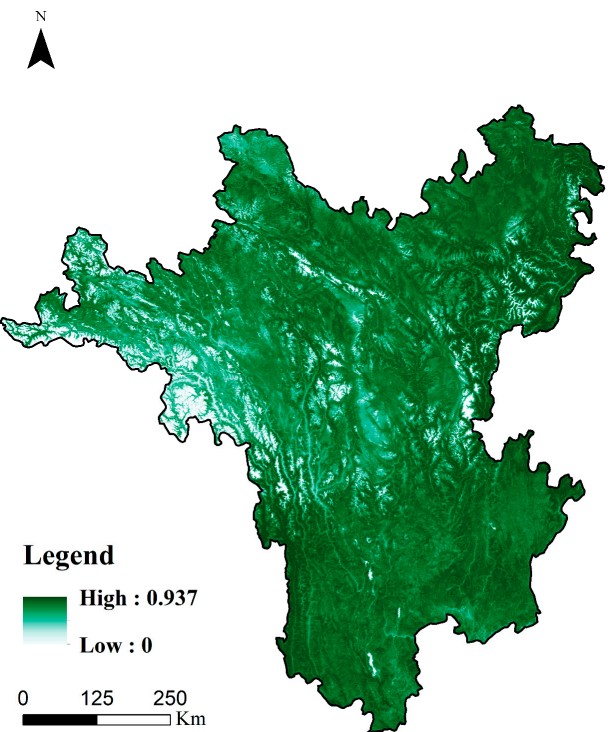

**Figure 10.** Spatial distribution of the NDVI (20-year average).

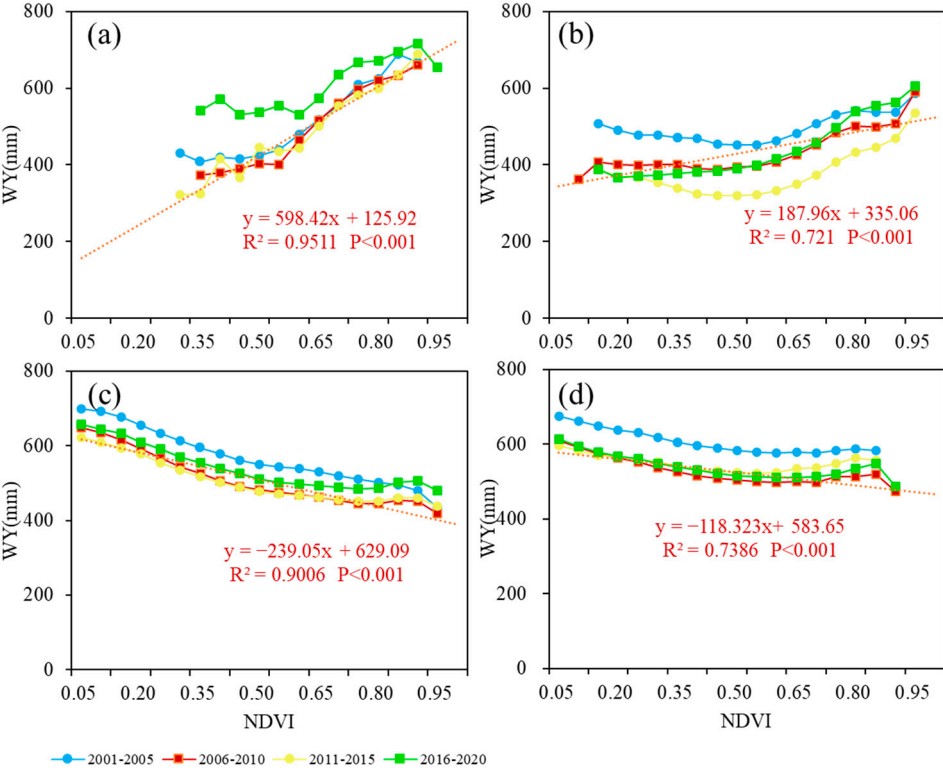

**Figure 11.** The relationships between the NDVI and WY at different elevation gradients: (**a–d**) are less than 1000 m, between 1000 m and 3500 m, between 3500 m and 5000 m, and more than 5000 m.

### 5.3. Factors Affecting Water Yield at Different Altitude Gradients

To further quantify the impacts of land use and climate on water yield at different elevation gradients, we used path analysis to explore the paths and strengths of the impacts between the LULC proportions (share of forested land, grassland, and unused land), climatic factors (PR, AET, LST), and WY in 2020 at different elevation gradients. Figure 12 shows the direct effect of each factor on WY, with PR and AET having the largest path coefficients (0.810 and −0.719, respectively) as factors directly affecting water yield, compared to LST, which has a non-significant direct effect on water yield, and LULC factors (the proportion of unutilized land, grassland, and forested land area) and the NDVI, which have a smaller direct effect on water yield. In addition, the LST and forest area proportion had a significant effect on AET, with path coefficients of 0.780 and 0.313, respectively, while the effect of precipitation on AET was not significant, which indicated that changes in the AET elevation gradient in the study area were mainly influenced by the LST and forest area proportion. In terms of indirect effects, the effects of the LST and forest land proportion on water yield were significantly significant (Table 5), with indirect path coefficients of −0.560 and −0.225, and total path coefficients of −0.660 and −0.293, respectively.

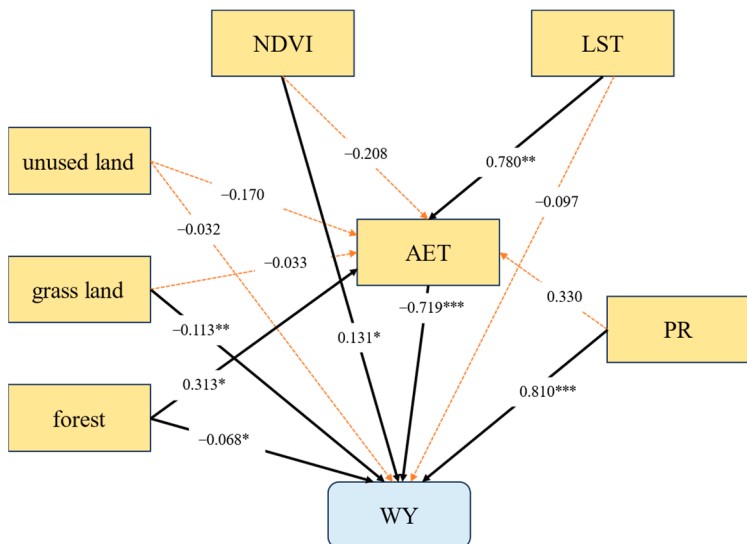

**Figure 12.** Path analysis diagram. Solid lines indicate significant impacts, dashed lines indicate non-significant impacts, numbers indicate path coefficients; *, **, and *** represent *p*-values less than 0.05, 0.01, and 0.001, respectively.

**Table 5.** Path coefficients of factors on water yield; *, **, and *** represent *p*-values less than 0.05, 0.01, and 0.001, respectively.

|  | LST | PR | Unused Land | Grassland | Forest | NDVI | AET |
|---|---|---|---|---|---|---|---|
| Direct path coefficient | −0.097 | 0.810 *** | −0.032 | −0.113 ** | −0.068 * | 0.131 * | −0.719 *** |
| Indirect path coefficient | −0.563 * | −0.237 | 0.122 | 0.024 | −0.225 * | 0.150 | - |
| Total path coefficient | −0.660 *** | 0.573 *** | 0.090 | −0.090 | −0.293 * | 0.280 | −0.719 *** |

The results showed that the differences in water yield across the altitudinal gradient were directly influenced by precipitation and actual evapotranspiration (ET), and the proportion of forest had a significant effect on WY by influencing the mediating variable AET. As AET at 2500–3500 m altitude is less constrained by water and energy, it is more suitable for forest growth [57] and has a higher area proportion. Whereas, due to the fact that the forest intercepts a large amount of radiation and consumes more water compared to other LULC types [58–60], it resulted in the study area having lower water yield above and below 3000 m elevation, which is consistent with the findings of others [61,62]. Many studies have

explained the variation in water yield with elevation as a result of the differences in the area of different land use types [27,36,63,64]. Although some studies pointed out the negative correlation between temperature and water yield at different elevation gradients [59], however, the results of the path analysis indicated that the indirect effect of temperature on water yield exceeded the effect of forest land occupation at the elevation gradient, which differed from the results of Dai et al.'s study in this region, which may be caused by the different study scales [35]. Studies have shown that AET may increase in mountainous regions in the context of global warming, leading to a decrease in water resources [24,65]. Although the drying trend has been reported to be slower at higher altitudes than at lower altitudes in the HDMR, this still poses some challenges for water resource management in mountainous regions.

*5.4. Uncertainty and Limitations*

The assessment of water yield services is a complex process, which entails uncertainties. The InVEST model, commonly used for this purpose, simplifies hydrological processes and disregards the influence of complex terrain, which includes the recharge of surface water and groundwater [66]. We opted to use TerraClimate, a gridded climate dataset, in our study, given its availability of data and the number of meteorological stations. The dataset was developed based on the WorldClim, CRU Ts4.0, and JRA-55 data [67]. However, it is worth noting that using TerraClimate may result in an overestimation of precipitation in mountainous areas [68,69]. Furthermore, the accuracy of our results may be affected by the LULC data sourced from the MCD12 product, as studies have shown that the problem of pixel mixing in complex mountainous areas due to the terrain and altitude increases the uncertainty of the product [70,71]. Additionally, the empirically determined KC coefficients and soil root depth in Table 2 may affect the accuracy of the results. Nevertheless, we expect the spatial distribution and basic pattern of the results to remain unchanged.

The HDMR, as a typical monsoon climate zone, experiences uneven annual precipitation distribution [72], with the dry season receiving less precipitation, and the rainy season (June-September) contributing 75–90% of the annual precipitation [73–75]. Our study focused on analyzing the interannual variation and spatial–temporal distribution of regional water yield. Given the seasonal characteristics of the HDMR climate, future research should concentrate on studying the changes in WY during different periods of the year to support regional water resources management.

## 6. Conclusions

The Hengduan Mountains, located in the upstream area of several rivers with complex topography and large altitude differences, are vital for water supply in China. This study utilized the InVEST annual water yield model to quantitatively assess the ecological service function of water yield in the Hengduan Mountains from 2001 to 2020 and identify the factors influencing it in relation to elevation. The findings are summarized below:

1.  The spatial pattern of water yield in the Hengduan Mountains for the past 20 years is consistent, showing a general decrease from southeast to northwest. For most of this 20-year period, the average annual water yield was concentrated between 300 mm and 700 mm, occupying about 95% of the area. The southwestern and eastern regions have high values of water yield, whereas the higher elevations in the northwestern area have low values.
2.  The water yield in the HDMR first decreased, reaching a minimum of 406 mm in 2011, and then increased from 2001 to 2020. It reached higher levels in 2004, 2018, and 2020. The water yield in the central and western HDMR decreased, whereas the eastern Sichuan Basin region showed an increase.
3.  The water yield services of the HDMR are affected by climate, vegetation, and elevation. Climatic factors are the primary influencing factors on the spatial and temporal variation of water yield in the area. Precipitation as the source of water yield is the main variable affecting the spatial and temporal patterns of water yield, and in most

areas, evapotranspiration and land surface temperature have a negative impact on water yield.

4.  Water yield varies greatly with altitudinal gradient, generally showing a decreasing and then increasing trend, with the lowest water yield at about 3000 m above sea level, which may be related to LULC at different altitudes. On the altitudinal gradient, precipitation and actual evapotranspiration had a high direct effect, and land surface temperature and forest proportion had a high indirect effect on water yield through actual evapotranspiration.

5.  The relationship between the NDVI and water yield is not a simple linear relation-ship and varies significantly with altitude. In the low and middle altitude regions, the two are positively correlated, while in the high-altitude region, they are negatively correlated.

**Author Contributions:** Conceptualization, Q.S., H.S. and L.H.; formal analysis, Q.S., L.H. and L.L.; funding acquisition, H.S., Q.S. and L.L.; investigation, Q.S., L.H., L.L. and J.Q.; methodology, Q.S., H.S. and L.H.; validation, H.S., Q.S. and L.H.; writing—original draft, Q.S., L.H. and H.S.; writing—review and editing, Q.S., L.L. and J.Q. All authors have read and agreed to the published version of the manuscript.

**Funding:** This study was funded by the National Natural Science Foundation of China (Grant No. 42271405) and the Science and Technology Department of Sichuan Province (Grant No. 2022NS-FSC0231, 2023NSFSC0248).

**Data Availability Statement:** The data presented in this study are contained within the article.

**Acknowledgments:** We are greatly grateful for the support of the funds and projects. We are also grateful for the anonymous reviewers and their insight and critical review of the manuscript.

**Conflicts of Interest:** The authors declare no conflict of interest.

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
