# Peer review of "Spatiotemporal Variation and Factors Influencing Water Yield Services in the Hengduan Mountains, China"

_remotesensing, doi:10.3390/rs15164087_

Round 1

Reviewer 1 Report

The paper remotesensing-2439767 analyzed the spatial and temporal variations of water yield and drivers. However, the innovation and significance of this article is unclear. Compared with similar studies, the authors found no new results, nor were the methods used novel. It can be seen that the authors are very careless. Chinese characters appear in the references and there are Chinese symbols in the text. The format of academic papers should be standardized.

Abstract: Please provide full words when using abbreviation for the first time.

Lines 38-41 water production or water yield?

The logic of the first three paragraphs of the Introduction is poor.

Line 68, The change of evapotranspiration is closely related to land use/vegetation change.

Many studies have been carried out on the impact of climate change and land use/land cover change on water yield and topographic effect. The author needs to collect more comprehensive literature in the introduction, summarize the methods and ideas of previous studies, and then summarize your innovations to support the significance of this research. Please well develop a hypothesis-based story by identifying the knowledge gaps via a recent literature review to demonstrate the novelty and importance of this study.

The authors used different resolution data, how to unify the resolution.

How are the parameters of the model set, whether it has been localized, it can be seen that many secondary types of land use/land cover are given the same parameters, then what is the reason for the author to use secondary classes?

The authors used the Budyko hypothesis in calculating water yield, which is often applied on multi-year time scales.

I looked up the InVEST model the authors used. This model has not been used correctly in many ecosystem service studies. The water yield model is based on a simple water balance where it is assumed that all water in excess of evaporative loss arrives at the outlet of the watershed. The model is an annual average time step simulation tool applied at the pixel level but reported at the subwatershed level. If possible, calibration of the model should be performed using long term average streamflow. As a rule of thumb, a 10-year period should be used to capture some climate variability, and this 10-year period should coincide with the date of the LULC map.

Is the runoff data used by the authors to verify water yield annual or multi-year averages?

As can be seen from Figure 3, the spatial pattern of water yield may change greatly.

An important question is whether extreme weather existed between 2001 and 2020.

Figure 4, p<0.1? Spatially, which areas of change are significant?

The analysis of the results is simple, only a time series analysis is done. What about terrain effects and drivers?

Table 4 needs to be redone.

The discussion is too simplistic and lacks depth, with no new findings compared to previous studies.

The authors did not pay attention to details, and Chinese words and symbols appeared.

Languages in some cases are difficult to understand and need revise.

Author Response

Thank you for your letter and for the reviewers’ comments concerning our manuscript entitled " Spatiotemporal Variation and factors influencing water yield services in the Hengduan Mountains, China" (Manuscript ID: remotesensing-2439767). The comments are very valuable and helpful for revising and improving the paper, as well as the important guiding significance to our researches. We carefully studied the review comments and made detailed revisions in the hope of approval. The red part of the original text is the modified content. The responses to the review comments are now provided in a word document.

Reviewer 2 Report

The manuscript focuses on the current hot issues, the value of ecosystem services, especially the water conservation capacity in the water source area, and makes use of more remote sensing data products in the data aspect, and carries out the water conservation calculation and analysis of the work, which is in line with the magazine on the subject. However, the manuscript has the following deficiencies, and I hope to improve to meet the requirements of publication in this journal. 1) References can be summarized in selected parts rather than described individually;

2) Precipitation products can be validated using precipitation from weather stations;

3) Are the parameters appropriate and are there any references for setting their values? e.g. root_depth and kc in Table 1

4) and is the resolution of the data real? and is the resolution of the data true? (the resolution of precipitation (PR) and reference evapotranspiration (ET0) are 1/24° in Table 2).

The presentation should be more concise and readable.

Author Response

(The authors gave the same response as above.)

Reviewer 3 Report

Line 27: what is PET, when first use the short format, suggest you give the full name.

Introduction Line 75-81: in this section, suggest you supply the methods used in existing studies, how others do this similar work? what methods they have used. This part is important.

Line 87-89: here you should add relevant references of vegetation destruction indeced by human activities. 

Section 2.2: suggest you rearrange these dataset into a table, introducing the data produced year, resolution, data type, accessed website, brief introduction, etc. In addition, why you use MCD12Q1 dataset to as LULC, this LULC dataset had raw accuracy, if you perform at Henduan mountain, suggest you use other LULC with high accuracy, based on my understanding, several LULC products had been published.

How do you treat these datasets with various resolution? resample? what resample method you have used? you should give detailed information.

The trend method you used Sens slope, however, this formula is not complete, in addition, MK test you should also done to analyze the trend situation of water yield.

Line 177: compared with actual total water volume for the eight cities, which cities? you should provide each cities' location and provide a spatial distribution map of these cities. In addition, the actual data? how you get these data? by statistical yearbook? if so, you should add them as data source in Table 2.

In addition, this part would be better in discussion part, here, suggest you divide the results and vertification into two separate parts, and named results, discussion. Then subtitle should be provided, for example, 3.1 the water yield spatial analysis of HDMR from 2001 to 2020. 3.2. ...

The results part is too short, you should rearrange your structure, some parts of discussion should be put into results part. Section 5.1&5.2 should be 3.2 3.3, the first paragraph of results part should be discussion part 5.1.

Generally, this work had some novelty, however, the structure and description need to modify. 

Author Response

Thank you for your letter and for the reviewers’ comments concerning our manuscript entitled " Spatiotemporal Variation and factors influencing water yield services in the Hengduan Mountains, China "(Manuscript ID: remotesensing-2439767). The comments are very valuable and helpful for revising and improving the paper, as well as the important guiding significance to our researches. We carefully studied the review comments and made detailed revisions in the hope of approval. The red part of the original text is the modified content. The responses to the review comments are now provided in a word document.

Reviewer 4 Report

This study analyzed the water production services and their influencing factors in the Hengduan Mountains of southwestern China, with detailed data and reliable results and conclusions. The following questions are worth referring to:

(1) In the Introduction, the first and third paragraphs, as well as the second and fourth paragraphs, seem to be interconnected. Please consider my immature suggestion.

(2) Suggest adding a website in a suitable location for the data source.

(3) Dai (2020) studied water production services in the Hengduan Mountains and suggested adding a comparison of similarities and differences in the results of you and Dai.

Dai, E., & Wang, Y. (2020). Attribution analysis for water yield service based on the geographical detector method: a case study of the Hengduan Mountain region. Journal of Geographical Sciences, 30, 1005-1020.

(4) Do the data in Figures 9 and 10 come from raster data? If so, how do they extract data?

(5) Some literature can be appropriately referenced:

Hu, W., Li, G., Gao, Z., Jia, G., Wang, Z., & Li, Y. (2020). Assessment of the impact of the Poplar Ecological Retreat Project on water conservation in the Dongting Lake wetland region using the InVEST model. Science of the Total Environment, 733, 139423.

Wei, P., Chen, S., Wu, M., Deng, Y., Xu, H., Jia, Y., & Liu, F. (2021). Using the InVEST model to assess the impacts of climate and land use changes on water yield in the upstream regions of the Shule River Basin. Water, 13(9), 1250.

Wang, X., Liu, G., Lin, D., Lin, Y., Lu, Y., Xiang, A., & Xiao, S. (2022). Water yield service influence by climate and land use change based on InVEST model in the monsoon hilly watershed in South China. Geomatics, Natural Hazards and Risk, 13(1), 2024-2048.

Author Response

(The authors gave the same response as above.)

Reviewer 5 Report

This study quantitatively evaluated the water volume in mountainous areas and analyzed the spatial and temporal changes in water volume from 2001 to 2020. The InVEST model was used and the influencing factors were examined in conjunction with altitude gradients. After review, the following problems need to be improved.

1.     This study investigated the impact of climate, vegetation, and land use changes on water conservancy services under different terrain conditions. The graphical gradient provides valuable insights for water safety management measures. Meteorological factors are directly influencing factors in water producing areas, while agricultural cultivation is a water consuming area. It is well known that these factors are the main driving factors for water production. How is this game relationship reflected in the research area of this article? What are the differences from other research areas?

2.     What is the resolution of the grid cells when using the Invest model? How are soil data with relevant resolutions, such as soil sand content, etc., obtained? Need detailed explanation. In addition, how are the model related parameters set? For example, empirical parameter Z, etc. detailed explanation is required.

3.The author has considered the impact of changes in water production in Hengduan Mountain from meteorological, land use, and other aspects, providing new ideas for the future allocation and management of water production in the research area. However, there are still certain limitations in the work. The driving factors considered should be more comprehensive, especially in mountainous areas. In reality, when the water production is affected by more factors, more consideration should be given to analyzing the influencing factors.

Minor problems

- Line 95, the analysis of the effect of climate on water production under different gradients is missing in the paper.

-Line 153, please explain how to get and set the z value in the InVEST model.

-Line 173, when verifying the water yield, please provide additional information on which year's water resources bulletin data is used for verification.

-Line 198, it is unreasonable to directly perform a linear fit of the total water production for each of the two time periods using 2011 as the boundary, and other fitting methods are recommended.

- Line 257, the spelling of "Poisson correlation coefficients" in the title of Figure 7 is wrong, please correct it.

- Line 275, the font format of Table 4 needs to be modified.

- Line 276, Figure 9(b) Lack of coordinate axes markings and missing legends.

- Line 291, the explanation of the vegetation cover of unused land and the water demand and evapotranspiration capacity of the forest is suggested to be added literature to the paper.

- The spatial analysis is conducted by administrative districts several times in the text, but the description of the location of administrative districts is missing in the text, please add additional descriptions of administrative districts.

This study quantitatively evaluated the water volume in mountainous areas and analyzed the spatial and temporal changes in water volume from 2001 to 2020. The InVEST model was used and the influencing factors were examined in conjunction with altitude gradients. After review, the following problems need to be improved.

1.     This study investigated the impact of climate, vegetation, and land use changes on water conservancy services under different terrain conditions. The graphical gradient provides valuable insights for water safety management measures. Meteorological factors are directly influencing factors in water producing areas, while agricultural cultivation is a water consuming area. It is well known that these factors are the main driving factors for water production. How is this game relationship reflected in the research area of this article? What are the differences from other research areas?

2.     What is the resolution of the grid cells when using the Invest model? How are soil data with relevant resolutions, such as soil sand content, etc., obtained? Need detailed explanation. In addition, how are the model related parameters set? For example, empirical parameter Z, etc. detailed explanation is required.

3.The author has considered the impact of changes in water production in Hengduan Mountain from meteorological, land use, and other aspects, providing new ideas for the future allocation and management of water production in the research area. However, there are still certain limitations in the work. The driving factors considered should be more comprehensive, especially in mountainous areas. In reality, when the water production is affected by more factors, more consideration should be given to analyzing the influencing factors.

Minor problems

- Line 95, the analysis of the effect of climate on water production under different gradients is missing in the paper.

-Line 153, please explain how to get and set the z value in the InVEST model.

-Line 173, when verifying the water yield, please provide additional information on which year's water resources bulletin data is used for verification.

-Line 198, it is unreasonable to directly perform a linear fit of the total water production for each of the two time periods using 2011 as the boundary, and other fitting methods are recommended.

- Line 257, the spelling of "Poisson correlation coefficients" in the title of Figure 7 is wrong, please correct it.

- Line 275, the font format of Table 4 needs to be modified.

- Line 276, Figure 9(b) Lack of coordinate axes markings and missing legends.

- Line 291, the explanation of the vegetation cover of unused land and the water demand and evapotranspiration capacity of the forest is suggested to be added literature to the paper.

- The spatial analysis is conducted by administrative districts several times in the text, but the description of the location of administrative districts is missing in the text, please add additional descriptions of administrative districts.

Author Response

(The authors gave the same response as above.)

Round 2

Reviewer 1 Report

I'm glad the authors tried their best to revise their manuscript. The authors are expected to examine the references carefully, both for content and format.

English is ok.

Author Response

Dear Editors and Reviewers,

Thank you for your letter and for the reviewers’ comments concerning our manuscript entitled " Spatiotemporal Variation and factors influencing water yield services in the Hengduan Mountains, China" (Manuscript ID: remotesensing-2439767). We have carefully checked the content and formatting of the references and updated them. Hopefully it will be approved.

Best regards.

Reviewer 3 Report

Authors have addressed all my concerns.

Author Response

(The authors gave the same response as above.)
